# Attitudes toward COVID-19 Vaccination: Staff and Patient Perspectives at Six Health Facilities in Sierra Leone

**DOI:** 10.3390/vaccines11081385

**Published:** 2023-08-19

**Authors:** Stefanie A. Joseph, Jean Gregory Jerome, Foday Boima, Pierre Ricard Pognon, Donald Fejfar, Yusupha Dibba, Daniel Lavalie, Mohamed Bailor Barrie, Aramide Oteju, Mohamed Sheku, Mariama Mahmoud, Jusu Mattia, Dale A. Barnhart

**Affiliations:** 1Partners In Health (PIH), Boston, MA 02199, USA; donlukef@gmail.com (D.F.); dale_barnhart@hms.harvard.edu (D.A.B.); 2Partners In Health Sierra Leone (PIH-SL), Freetown, Sierra Leone; gjerome@pih.org (J.G.J.); fboima@pih.org (F.B.); rpognon@pih.org (P.R.P.); ydibba@pih.org (Y.D.); bbarrie@pih.org (M.B.B.); aoteju@pih.org (A.O.); mgsheku42@gmail.com (M.S.); 3Ministry of Health and Sanitation, Freetown, Sierra Leone; danlavalie1023@yahoo.com (D.L.); m_mahmoud85@yahoo.com (M.M.); jusumattia@yahoo.com (J.M.); 4Harvard Medical School, Boston, MA 02115, USA

**Keywords:** COVID-19, Sierra Leone, health care workers, vaccine hesitancy, vaccine intention, vaccine access

## Abstract

Sierra Leone is a West African country with a population of over 8 million. With more than half of Sierra Leone’s population living in rural areas, it is important to understand rural populations’ access to and attitudes toward the COVID-19 vaccine. In November 2021, the rate of vaccination coverage in Sierra Leone was only 7% for one dose and 4% for two doses. Understanding perspectives of health facility staff and patients can help strengthen future vaccine campaigns. We conducted a cross-sectional study, between March 2022 and May 2022, of clinical staff, non-clinical staff, and adult (>18 years) patients/caregivers attending six Ministry of Health and Sanitation (MoHS) facilities supported by Partners In Health, four in the Kono district and two in the Western Urban Area district, the capital of Sierra Leone. We assessed the opportunity to vaccinate, vaccine uptake, and intention to vaccinate. Out of the 2015 participants, 11.4% were clinical staff, 18.8% were non-clinical staff, and 69.8% were patients/caregivers. Less than half of the patients/caregivers had the opportunity to be vaccinated (42%), and 22% of patients/caregivers were fully vaccinated. Among the unvaccinated population, 44% would refuse a vaccine if offered to them at no cost. Lack of access to COVID-19 vaccines and to official education messaging, especially for patients and caregivers, is still an underlying problem in Sierra Leone for vaccine uptake, rather than a lack of willingness to be vaccinated.

## 1. Introduction

As of 24 May 2023, there have been 766,895,075 cases and 6,935,889 deaths globally due to COVID-19 [1]. Sierra Leone recorded its first case/emergency in March 2020, with 7762 cases, 125 deaths, and 7,548,308 vaccine doses administered since [2], although these statistics may be underreported due to a lack of testing found in many low- and middle-income countries (LMICs). In Sierra Leone, many pre-existing factors have challenged the country’s ability to respond to the COVID-19 pandemic, including limited access to health care and preventative public health measures in rural areas [3].

While the COVID-19 vaccine has proven to be an effective tool for controlling the pandemic and preventing severe illness [4], this rollout was initially more accessible for high-income countries where strong economic conditions and high levels of education expedite the process [5,6]. COVID-19 vaccination campaigns could bolster the ability of health systems in Sierra Leone and other LMICs to curb the pandemic. To date, a lack of financing and equal access to vaccines among LMICs have limited the availability of COVID-19 vaccines in Sierra Leone [7,8,9,10]. However, trust in vaccines as well as the institutions that administer them is a key determinant of the success of any vaccination campaign [11]. Consequently, as COVID-19 vaccines become more available in Sierra Leone, vaccine uptake and intention to vaccinate will become key factors that will dictate the effectiveness of vaccination campaigns.

Although COVID-19 vaccines have been demonstrated to be extremely effective [4], the poor uptake of vaccines in high-income settings has already become a major barrier to the global COVID-19 response [11,12,13]. Initial studies suggested that select low-income countries had similar attitudes toward COVID-19 vaccines as high-income settings, but vaccine hesitancy is a complex and context-specific phenomenon that evolves [14,15,16]. As COVID-19 vaccines become more widely available in LMICs, additional research is needed to understand context-specific factors that may affect uptake, especially research that is focused on rural areas where the population may have less access to education, mass media, and health care [17]. To date, few studies have assessed vaccine hesitancy in Sierra Leone [18,19,20]. To better inform ongoing vaccination campaigns, including communications and outreach efforts in Sierra Leone, this study aimed to assess the current attitudes and perceptions of COVID-19 vaccines for individuals living in the Kono district and Western Urban Area district, Sierra Leone.

## 2. Materials and Methods

### 2.1. Study Setting

Sierra Leone is a West African country that is among the poorest in the world, with about 57% of its more than 8 million residents living below the poverty line today [21,22]. Its health system is recovering after two major crises — an 11-year civil war and the Ebola epidemic — while also growing in order to reduce the overall burden of maternal mortality, malnutrition, infectious diseases, non-communicable diseases, and chronic conditions [23,24]. Partners In Health (PIH) is a non-profit organization that works with the Ministry of Health and Sanitation (MoHS) on health systems to strengthen activities in the Kono district and Western Urban Area district, Sierra Leone. Activities include the provision of direct clinical care, clinical and community health worker training, supply chain system support, and innovative clinical interventions to improve patient outcomes and build health care delivery systems. Currently, PIH supports health care services in the Kono district (Koidu Government Hospital, Wellbody Clinic, Sewafe Community Health Center, and Kombayendeh Community Health Center) and in the Western Urban Area district (Lakka Government Hospital and Kissy Psychiatric Teaching Hospital) in Freetown.

This study was conducted in six MoHS health facilities. Four facilities (Koidu Government Hospital, Wellbody Clinic, Sewafe Community Health Center, and Kombayendeh Community Health Center) are in the Kono district, an eastern region district with a catchment population of 506,100 people [25], 75% of whom live in rural areas [26]. Two facilities (Lakka Government Hospital and Kissy Psychiatric Teaching Hospital) are in the Western Urban Area district in Freetown, Sierra Leone, a major urban, economic, cultural, educational, and political center with a population of more than one million people [25]. PIH has been supporting MoHS services in these facilities since 2015.

### 2.2. Study Design and Study Population

A convenience sampling method was used to gather data on as many participants as available during the study collection period. We recruited adult patients over 18 years of age or accompanying adult caregivers who were waiting in the various outpatient departments. Any patient who was in medical distress or not mentally capable of providing consent was excluded from the study.

Clinical and non-clinical staff in facilities were also eligible to participate. In order to estimate the proportion of respondents who would be willing to be vaccinated with a 95% confidence interval of +/−3%, we estimated our target sample size should be 1000. This sample size calculation assumes that the proportion of respondents who are willing to be vaccinated is similar to what has been previously observed in PIH-supported populations in Haiti and Malawi (approximately 25%). However, depending on respondent availability, we anticipated that actual enrollment could range from 500 (precision of +/−4%) to 2500 individuals (precision of +/−2%).

### 2.3. Data Collection

Data was collected from 29 March 2022 to 11 May 2022. Enumerators fluent in Krio verbally administered a 15-to-25-minute questionnaire in Krio focused on vaccination status, intention to vaccinate, attitudes toward the COVID-19 vaccine, and sources of information about the COVID-19 vaccine. Non-identifiable demographic characteristics, such as age, sex, ethnicity, education, facility location, and role at the facility, were also collected. Data was collected electronically using tablets and the CommCare app.

All study staff that engaged in data collection received a two-day training in research ethics, including respect for study participants, consent procedures, and secure storage and maintenance of data before the start of the evaluations. Additionally, they received survey-specific training during the study period.

### 2.4. Data Analysis

Vaccine uptake was defined as whether or not a person received a COVID-19 vaccine and reflected vaccine access along with willingness to be vaccinated [27]. Vaccination status was broken down into multiple categories: (1) opportunity to be vaccinated defined as ever having the opportunity to receive a vaccine for COVID-19; (2) received vaccine or partially vaccinated meaning receiving at least one dose of a COVID-19 vaccine; (3) fully vaccinated as in receiving both doses of a COVID-19 vaccine or one dose if the vaccine manufacturer was Johnson & Johnson; and (4) not vaccinated as in not having the opportunity to be vaccinated or refused to receive the vaccine. Intention to vaccinate was defined as whether or not a person would receive a vaccine if it were made available to them today at no cost [27].

Additionally, we assessed attitudes of COVID-19 by intention to vaccinate, trusted sources of COVID-19 information, and quality of care at the vaccination site. Responses were compared among vaccination statuses. Participants reported their trust in a source using a 5-point Likert Scale, ranging from “strongly distrust” to “trust a lot” options. All survey questions are available in Appendix A.

Participants were asked open-ended questions as to why they refused vaccination, reasons they would or would not be willing to receive the COVID-19 vaccine, and their primary sources of COVID-19 information. After hearing their responses, enumerators would code them into synonymous pre-defined categories. If a response did not fit into one of these categories, there was an option to select “Other” and provide a write-in response. Categories were developed based on previous survey tools on attitudes toward the COVID-19 vaccine [11,15,27,28]. Reasons for and against vaccination were observed among clinicians, non-clinicians, outpatients, and caregivers.

Data was analyzed using STATA 15.1 [29]. The statistical analysis consisted primarily of descriptive statistics, with data summarized using frequencies, percentages, and bar graphs. Chi-squared or Fisher’s exact tests were used to assess the associations between demographic characteristics and outcomes of interest. Univariate and multivariate logistic regressions were conducted to assess predictors of low access (having no opportunity to be vaccinated) and predictors of non-acceptance (responded no or unsure to the intention of receiving a COVID-19 vaccine). Backward stepwise regression was used to eliminate variables from the full model. The full model was reduced until all demographic variables had a *p* < 0.05.

### 2.5. Ethical Considerations

The study obtained ethical approval on 27 January 2022 from the PIH Sierra Leone Research and Ethic Committee and the Sierra Leone Ethics and Scientific Review Committee, IRB# 1.0, before the start of data collection. All data was stored securely and anonymously.

## 3. Results

A total of 2015 participants completed the questionnaire, 230 (11.4%) clinicians, 379 (18.8%) non-clinical staff, and 1406 (69.8%) outpatients or patient caregivers (Table 1). A majority of the study population was enrolled at the Koidu Government Hospital (30.6%) or Wellbody Clinic (25.9%), 51.6% were women, and 1736 participants (86.2%) were less than 45 years old. More than half of the respondents completed education higher than primary school, with 81.3% of clinicians completing post-secondary education and 33.1% of outpatients or patient caregivers having received no formal education. Individuals mostly identified with being Kono (n = 771, 38.3%) or Krio (n = 562, 27.9%). All three groups cited that mass media was their primary source of COVID-19 vaccine information, with clinicians and staff being more likely to cite social media, the national government, or the Ministry of Health (MoH) as a primary source, while outpatients or patients’ caregivers chose their family or friends as a primary source. Location of the study interview, sex, age, level of education, identifying as Kono, Mende, or Mandingo, and all primary sources of COVID-19 vaccine information were significantly associated (*p* < 0.05) with the respondents’ role at the facility.

Figure 1a shows a bar graph of COVID-19 vaccine uptake by facility role. Clinical (n = 183, 80%) and non-clinical staff (n = 254, 67%) reported having more opportunities to receive the vaccine compared to outpatients/patient caregivers (n = 592, 42%), *p* < 0.001. Most staff members were fully vaccinated; however, only 22% of all outpatients/caregivers received complete dosages. Figure 1b presents a bar graph of the intention to vaccinate by facility role. Overall, the intention to vaccinate was high in all three groups, although it was the highest among clinicians (82%) and lowest among the patients/caregivers (70%).

Actual vaccination status was associated with willingness to be vaccinated in a hypothetical situation where a vaccine was offered today at no cost (Table 2). Over 95% of those who were fully vaccinated would say “Yes” to an approved COVID-19 vaccine if it were offered to them today at no cost. Among those who were not vaccinated, only 53.8% were willing to be vaccinated. Out of all the vaccine types listed, individuals were more willing to receive AstraZeneca (55.9%) or Johnson & Johnson (55.3%). In general, participants chose the MoHS, local leaders, and family or friends as trusted sources to relay information about the COVID-19 vaccine. However, the vaccinated and partially vaccinated were more likely to cite these sources than the not vaccinated. In contrast, the not vaccinated were more likely to cite social media as a primary source. When participants discussed the quality of care during vaccination, partially vaccinated people reported a lower quality of care compared to those who were fully vaccinated.

Figure 2 presents reasons for and against vaccination among facility roles. More than 90% of all facility roles reported “For my health” as a reason for vaccination. A high proportion of staff supported COVID-19 vaccination recommendations by the government and mandates by their workplaces. Perceived benefits such as “For the health of my family” and “For the health of my community” were more commonly cited among outpatients and patient caregivers than clinical and non-clinical staff. Among those who would refuse or were unsure about a COVID-19 vaccine, more than 60% reported “Personally not at risk” and “Distrusts vaccine manufacturers” as reasons against vaccination. Compared to staff, outpatients and patient caregivers were significantly more likely to report “COVID-19 is not dangerous”, “Vaccine was designed to harm me”, “Religion beliefs”, and “Concerns about specific vaccine types” as reasons for being unwilling to receive the COVID-19 vaccine.

Additionally, we reported predictors of low access to COVID-19 vaccines (Table 3). About 49% of participants reported having no opportunity to be vaccinated. There were meaningful differences (*p* < 0.05) between those who had an opportunity to be vaccinated and those who did not in terms of location, role at the facility, sex, age, level of education, and all primary sources of COVID-19 information. Significant differences in vaccine accessibility could also be observed among those who identified as Kono, Krio, Temne, Limba, Fullah, and Loko. In the multivariate analysis, better vaccine access could be seen among those who identified as Kono (OR: 0.71; 95% CI; 0.56–0.91; *p* = 0.006) and received COVID-19 information from mass media (OR: 0.71; 95% CI: 0.54–0.92; *p* = 0.011), social media (OR: 0.74; 95% CI: 0.58–0.93; *p* = 0.011), and health care workers (OR: 0.44; 95% CI: 0.34–0.56; *p* < 0.001). Key predictors of low vaccine access included participating at Kissy Psychiatric Teaching Hospital (OR: 8.37; 95% CI: 4.89–14.33; *p* < 0.001) or Lakka Government Hospital (OR: 6.45; 95% CI:3.76–11.06; *p* < 0.001), being an outpatient or patient caregiver (OR: 4.21; 95% CI: 2.62–6.77; *p* < 0.001), being younger (OR: 4.62; 95% CI: 3.09–6.93; *p* < 0.001), and having a lower level of education (OR: 3.00; 95% CI: 1.98–4.55; *p* < 0.001).

Lastly, we analyzed the predictors of vaccine hesitancy (Table 4). Five hundred and thirty-five participants (27.1%) reported “No” or “Unsure” when presented with a hypothetical situation where a vaccine was offered today at no cost and were considered vaccine hesitant. In our univariate analysis, the following demographics were significantly different when comparing those who did and did not intend to be vaccinated: location, level of education, identifying as Kono, Krio, Temne, or Fullah, and receiving their primary source of COVID-19 information from mass media, family or friends, the national government or MoH, social media, health care workers, or an employer. In the multivariate analysis, predictors of vaccine hesitancy included participating at Kissy Psychiatric Teaching Hospital (OR: 32.90; 95% CI: 13.78–78.58; *p* < 0.001) or Lakka Government Hospital (OR: 26.37; 95% CI: 10.77–64.58; *p* < 0.001), being younger (OR: 1.98; 95% CI: 1.29–3.06; *p* = 0.009), having a lower level of education (OR: 3.16; 95% CI: 2.18–4.57; *p* < 0.001), and identifying as Krio (OR: 2.97; 95% CI: 2.20–4.01; *p* < 0.001). Individuals who reported obtaining COVID-19 information from mass media (OR: 0.65; 95% CI: 0.50–0.85; *p* = 0.002) and health care workers (OR: 0.31; 95% CI: 0.23–0.42; *p* < 0.001) had lower odds of vaccine hesitancy.

## 4. Discussion

Overall, our study found low access to vaccines but high willingness to be vaccinated among patients and caregivers at hospitals in Sierra Leone. Only 42% of patients/caregivers reported having previous opportunities to be vaccinated, and 22% were fully vaccinated. However, 70% of patients and caregivers and 54% of the currently unvaccinated were willing to be vaccinated if a vaccine were provided to them for free. This level of vaccine acceptance is much higher than what was previously reported in a systematic review of African studies showing a vaccine acceptance rate of 49% [30]. There are various possible explanations for this finding. First, the high level of intention to be vaccinated may be associated with the post-Ebola outbreak context in Sierra Leone, leading people to be more receptive to public health messages about disease outbreaks than other populations. Second, a preparedness strategy against an Ebola outbreak at the Guinea borders with the Kono district coincided with the vaccine campaign for COVID-19, and this could have resulted in willingness to be vaccinated being substantially higher in the Kono district than in other hospitals. As a result, and due to the fear associated with the resurgence of an Ebola outbreak, respondents may have had this in mind when responding about their attitudes toward COVID-19 vaccines. Additionally, due to the severity and urgency of patients’ conditions at specialty hospitals, vaccination may not be the highest priority, therefore demonstrating poor vaccine outcomes at Lakka Government Hospital and Kissy Psychiatric Teaching Hospital.

Unsurprisingly, our data showed significant differences between respondents with a facility role (clinical and non-clinical staff) compared to patients/caregivers. Respondents with facility roles were more likely to have had the opportunity to receive the vaccine, to be fully vaccinated, and to exhibit a willingness to be vaccinated in the future. This difference may have been associated mainly with the adopted strategy to prioritize people with facility roles during the first rounds of COVID-19 vaccine campaigns, as they were considered to be of higher risk [30]. This may also be explained by the fact that respondents with a facility role were more likely to have a higher level of education. Lower access to COVID-19 vaccines among younger populations could be similarly explained by a strategy to prioritize the elderly.

High intention to vaccinate among health care workers (82%) may be due to attending both PIH and MoHS sensitization sessions at the beginning of the pandemic, where they were educated about the COVID-19 vaccine and disease. Patients and caregivers, on the other hand, were not offered the same opportunities to gain information or lessen their concerns. Intention to vaccinate may also be high due to the Ebola outbreak in Sierra Leone, inclining people to listen to public health messages about disease outbreaks and follow preventive measures to protect themselves and their family. This claim was further supported through our study, as 90% of facility roles mentioned the protection of their own health as a reason to accept the COVID-19 vaccine, and greater than 65% of clinical and non-clinical staff listed recommendations by the government as a reason for vaccine intent. The study also showed that the MoHS, local leaders, and family or friends are generally the most trusted among those fully and partially vaccinated or even unvaccinated, showing how people have built confidence in their leaders to make the right decisions to protect their health and that of their family and their community.

A limitation of our study was that it was only conducted in MoHS facilities supported by PIH, where this organization facilitates COVID-19 vaccine campaigns and messaging. PIH has also been involved in holding sensitization sessions with staff, resulting in positive change in staff perception and attitudes toward the COVID-19 vaccine. Future studies can compare PIH-supported MoHS facilities to non-PIH-supported MoHS facilities to understand how PIH’s efforts impacted our findings.

The study design also included a convenience sampling method; therefore, our study population may not be representative of the general public. Being that the study took place within health care facilities, social desirability bias may also play a role, i.e., participants could have answered in a manner that is favorable toward facility regulations.

Despite these limitations, our study points to several potentially modifiable strategies that could be used to improve the access to and uptake of vaccines. These include assigning community members or health facility staff as ambassadors to promote vaccination, increasing vaccine availability in specialty and teaching hospitals, creating outreach materials to be accessible in multiple languages, informing individuals throughout the vaccine process (type received, timing of second dose), and spreading educational messages through social media.

## 5. Conclusions

In the settings where the study was conducted in Sierra Leone, despite low availability of the vaccines for the targeted population, vaccine hesitancy remained relatively low among the respondents, especially among the respondents who were staff members at the facilities studied. This indicated that the lack of access to vaccines and to official education messaging, rather than lack of willingness to be vaccinated, is the primary barrier to successful COVID-19 vaccine campaigns in Sierra Leone. Therefore, efforts should be concentrated around tackling these issues to reach populations beyond health facility staff.

## Figures and Tables

**Figure 1 vaccines-11-01385-f001:**
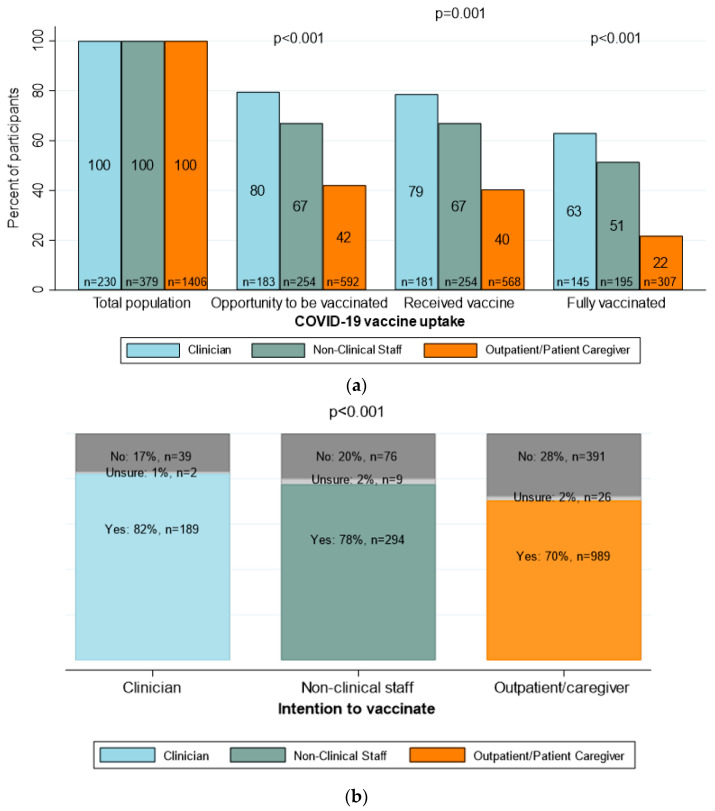
(**a**) COVID-19 vaccine uptake by facility role (N = 2015); (**b**) intention to vaccinate by facility role (N = 2015).

**Figure 2 vaccines-11-01385-f002:**
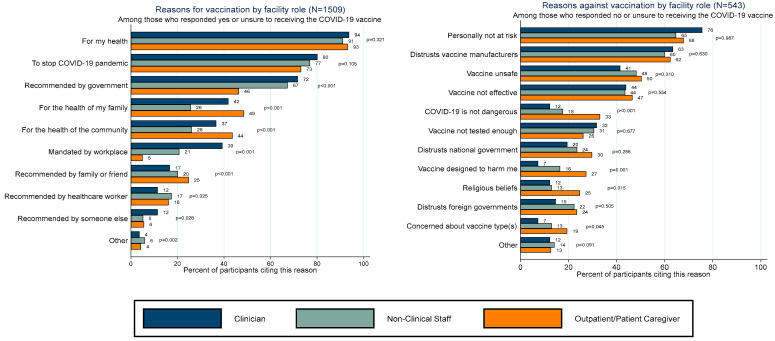
Reasons for and against the intention to vaccinate by facility role (N = 2015).

**Table 1 vaccines-11-01385-t001:** Respondent characteristics by facility role (N = 2015).

	Total	Clinician	Non-Clinical Staff	Outpatient/Patient Caregiver	*p*-Value
	N = 2015	N = 230	N = 379	N = 1406	
**Location**					<0.001
Koidu Government Hospital	617 (30.6%)	71 (30.9%)	139 (36.7%)	407 (28.9%)	
Wellbody Clinic	522 (25.9%)	39 (17.0%)	99 (26.1%)	384 (27.3%)	
Sewafe Community Health Center	140 (6.9%)	13 (5.7%)	4 (1.1%)	123 (8.7%)	
Konbayedeh Community Health Center	115 (5.7%)	5 (2.2%)	16 (4.2%)	94 (6.7%)	
Lakka Government Hospital	287 (14.2%)	50 (21.7%)	61 (16.1%)	176 (12.5%)	
Kissy Psychiatric Teaching Hospital	334 (16.6%)	52 (22.6%)	60 (15.8%)	222 (15.8%)	
**Sex (N = 1994)**					<0.001
Female	1028 (51.6%)	143 (62.2%)	107 (28.5%)	778 (56.0%)	
Male	966 (48.4%)	87 (37.8%)	268 (71.5%)	611 (44.0%)	
**Age, categorized (N = 2013)**					<0.001
18–24	401 (19.9%)	19 (8.3%)	40 (10.6%)	342 (24.4%)	
25–34	817 (40.6%)	94 (40.9%)	173 (45.6%)	550 (39.2%)	
35–44	518 (25.7%)	83 (36.1%)	111 (29.3%)	324 (23.1%)	
45–54	198 (9.8%)	23 (10.0%)	43 (11.3%)	132 (9.4%)	
55–64	61 (3.0%)	9 (3.9%)	11 (2.9%)	41 (2.9%)	
65+	18 (0.9%)	2 (0.9%)	1 (0.3%)	15 (1.1%)	
**Level of education (N = 2002)**					<0.001
None	519 (25.9%)	1 (0.4%)	56 (14.8%)	462 (33.1%)	
Some primary	90 (4.5%)	0 (0.0%)	16 (4.2%)	74 (5.3%)	
Complete primary	84 (4.2%)	1 (0.4%)	9 (2.4%)	74 (5.3%)	
Some secondary	540 (27.0%)	2 (0.9%)	105 (27.8%)	433 (31.1%)	
Complete secondary	402 (20.1%)	39 (17.0%)	118 (31.2%)	245 (17.6%)	
Post-secondary	367 (18.3%)	187 (81.3%)	74 (19.6%)	106 (7.6%)	
**Ethnicity ^1^**					
Kono	771 (38.3%)	73 (31.7%)	135 (35.6%)	563 (40.0%)	0.027
Krio	562 (27.9%)	53 (23.0%)	98 (25.9%)	411 (29.2%)	0.092
Mende	355 (17.6%)	89 (38.7%)	91 (24.0%)	175 (12.4%)	<0.001
Temne	266 (13.2%)	32 (13.9%)	52 (13.7%)	182 (12.9%)	0.880
Limba	196 (9.7%)	14 (6.1%)	44 (11.6%)	138 (9.8%)	0.080
Fullah	162 (8.0%)	7 (3.0%)	12 (3.2%)	143 (10.2%)	<0.001
Mandingo	162 (8.0%)	19 (8.3%)	34 (9.0%)	109 (7.8%)	0.730
Other	104 (5.2%)	11 (4.8%)	24 (6.3%)	69 (4.9%)	0.520
Loko	87 (4.3%)	11 (4.8%)	17 (4.5%)	59 (4.2%)	0.910
Korankoh	71 (3.5%)	8 (3.5%)	9 (2.4%)	54 (3.8%)	0.390
Sherbro	38 (1.9%)	6 (2.6%)	11 (2.9%)	21 (1.5%)	0.140
**Primary source of COVID-19 vaccine information ^1^**					
Mass media	1527 (75.8%)	194 (84.3%)	302 (79.7%)	1031 (73.3%)	<0.001
Family or friends	1037 (51.5%)	88 (38.3%)	188 (49.6%)	761 (54.1%)	<0.001
National government/Ministry of Health	898 (44.6%)	127 (55.2%)	232 (61.2%)	539 (38.3%)	<0.001
Social media	835 (41.4%)	128 (55.7%)	220 (58.0%)	487 (34.6%)	<0.001
Health care workers	564 (28.0%)	78 (33.9%)	136 (35.9%)	350 (24.9%)	<0.001
Local leaders	295 (14.6%)	17 (7.4%)	50 (13.2%)	228 (16.2%)	0.002
Employer	85 (4.2%)	19 (8.3%)	37 (9.8%)	29 (2.1%)	<0.001

^1^ Respondents could indicate multiple ethnicities and primary sources of information about COVID-19 vaccines.

**Table 2 vaccines-11-01385-t002:** Uptake and attitudes toward vaccines by COVID-19 vaccination status (N = 2015).

		COVID-19 Vaccination Status			
	Total	Fully Vaccinated	Partially Vaccinated	Not Vaccinated ^1^	*p*-Value
	N = 2015	N = 647	N = 356	N = 1012	
**Intention to vaccinate against COVID-19**					
If an approved vaccine to prevent COVID-19 was available to you today at no cost					<0.001
No	506 (25.1%)	25 (3.9%)	36 (10.1%)	445 (44.0%)	
Yes	1472 (73.1%)	616 (95.2%)	312 (87.6%)	544 (53.8%)	
Unsure	37 (1.8%)	6 (0.9%)	8 (2.2%)	23 (2.3%)	
**Which of the following COVID-19 vaccines would you be willing to receive?**					
AstraZeneca/Oxford (N = 1499)	1127 (55.9%)	412 (63.7%)	252 (70.8%)	463 (45.8%)	<0.001
Johnson & Johnson (N = 1499)	1114 (55.3%)	426 (65.8%)	220 (61.8%)	468 (46.2%)	<0.001
Sinopharm/Chinese National Biotec Group (N = 1499)	1033 (51.3%)	353 (54.6%)	233 (65.4%)	447 (44.2%)	<0.001
Pfizer/BioNTech (N = 1499)	900 (44.7%)	283 (43.7%)	200 (56.2%)	417 (41.2%)	<0.001
Moderna (N = 1499)	896 (44.5%)	281 (43.4%)	198 (55.6%)	417 (41.2%)	<0.001
None of these (N = 605)	33 (1.6%)	11 (1.7%)	7 (2.0%)	15 (1.5%)	0.007
**Trusted sources of information about the COVID-19 vaccine ^2^**					
The Ministry of Health (N = 2008)	1339 (66.7%)	508 (78.9%)	274 (77.2%)	557 (55.2%)	<0.001
Local leaders (N = 2008)	1339 (66.7%)	508 (78.9%)	274 (77.2%)	557 (55.2%)	<0.001
Family or friends (N = 2008)	1339 (66.7%)	508 (78.9%)	274 (77.2%)	557 (55.2%)	<0.001
Facility-based healthcare worker (N = 1938)	1243 (64.1%)	466 (74.7%)	261 (75.4%)	516 (53.3%)	<0.001
World Health Organization (N = 2011)	1304 (64.8%)	480 (74.5%)	272 (76.6%)	552 (54.5%)	<0.001
Community health workers (N = 1991)	1187 (59.6%)	433 (67.6%)	256 (73.4%)	498 (49.8%)	<0.001
Regional health authorities (N = 1977)	1119 (56.6%)	395 (61.6%)	245 (70.2%)	479 (48.5%)	<0.001
Mass media (N = 1924)	620 (32.2%)	179 (29.1%)	111 (32.8%)	330 (34.0%)	0.120
Social media (N = 1985)	453 (22.8%)	99 (15.5%)	81 (22.9%)	273 (27.5%)	<0.001
**Quality of care during COVID-19 vaccination among fully or paritally vaccinated (N = 1003)**					
**Did anyone tell you which type of vaccine you would receive? (N = 1002)**	486 (48.5%)	349 (54.0%)	137 (38.5%)	N/A	<0.001
**Did anyone tell you the timing for your second dose? (N = 996)**	518 (52.0%)	343 (53.6%)	175 (49.2%)	N/A	0.180
**Did anyone tell you about possible adverse events following immunization? (N = 989)**	593 (60.0%)	397 (62.2%)	196 (55.8%)	N/A	0.050
**Were COVID-19 prevention measures maintained at the vaccination site? (N = 991)**	658 (66.4%)	435 (68.3%)	223 (63.0%)	N/A	0.091
**How satisfied were you with COVID-19 vaccination process? (N = 1001)**					<0.001
Very dissatisfied	9 (0.9%)	5 (0.8%)	4 (1.1%)	N/A	
Dissatisfied	19 (1.9%)	4 (0.6%)	15 (4.2%)	N/A	
Neutral	144 (14.4%)	76 (11.8%)	68 (19.2%)	N/A	
Satisfied	603 (60.2%)	428 (66.3%)	175 (49.3%)	N/A	
Very satisfied	226 (22.6%)	133 (20.6%)	93 (26.2%)	N/A	

^1^ Includes both those who had no opportunity to receive the COVID-19 vaccine (N = 986) and those who refused the vaccine (N = 26). ^2^ Indicates proportion of patients who trust this source a lot.

**Table 3 vaccines-11-01385-t003:** Predictors of low access to COVID-19 vaccines (no opportunity to be vaccinated) (N = 1974).

	Univariate			Logistic Regression Model		
	No Opportunity	Had an Opportunity	*p*-Value ^2^	OR	95% CI	*p*-Value ^3^
	N = 967	N = 1007				
**Location**			<0.001			<0.001
Koidu Government Hospital	230 (23.8%)	369 (36.6%)		1.36	0.85–2.20	
Wellbody Clinic	158 (16.3%)	347 (34.5%)		1.11	0.68–1.82	
Sewafe Community Health Center	87 (9.0%)	53 (5.3%)		1.66	0.93–2.94	
Konbayedeh Community Health Center	54 (5.6%)	60 (6.0%)		1.00	---	
Lakka Government Hospital	194 (20.1%)	92 (9.1%)		6.45	3.76–11.06	
Kissy Psychiatric Teaching Hospital	244 (25.2%)	86 (8.5%)		8.37	4.89–14.33	
**Role at the facility**			<0.001			<0.001
Clinician	47 (4.9%)	182 (18.1%)		1.00	---	
Non-clinical staff	123 (12.7%)	250 (24.8%)		1.72	1.04–2.86	
Outpatient/patient caregiver	797 (82.4%)	575 (57.1%)		4.21	2.62–6.77	
**Sex**			0.001			---
Female	534 (55.2%)	482 (47.9%)		---	---	
Male	433 (44.8%)	525 (52.1%)		---	---	
**Age, categorized (years)**			<0.001			<0.001
18–24	234 (24.2%)	158 (15.7%)		4.62	3.09–6.93	
25–34	396 (41.0%)	409 (40.6%)		2.66	1.88–3.76	
35–44	229 (23.7%)	276 (27.4%)		1.56	1.09–2.24	
≥45	108 (11.2%)	164 (16.3%)		1.00	---	
**Level of education**			<0.001			<0.001
Did not complete primary	370 (38.3%)	232 (23.0%)		3.00	1.98–4.55	
Completed primary	344 (35.6%)	270 (26.8%)		2.02	1.35–3.02	
Completed secondary	146 (15.1%)	248 (24.6%)		1.27	0.84–1.93	
Post-secondary	107 (11.1%)	257 (25.5%)		1.00	---	
**Ethnicity ^1^**						
Kono	280 (29.0%)	477 (47.4%)	<0.001	0.71	0.56–0.91	0.006
Krio	304 (31.4%)	253 (25.1%)	0.002	---	---	---
Mende	155 (16.0%)	194 (19.3%)	0.060	---	---	---
Temne	168 (17.4%)	97 (9.6%)	<0.001	---	---	---
Limba	108 (11.2%)	80 (7.9%)	0.015	---	---	---
Fullah	94 (9.7%)	66 (6.6%)	0.010	---	---	---
Mandingo	69 (7.1%)	90 (8.9%)	0.140	---	---	---
Other	49 (5.1%)	54 (5.4%)	0.770	---	---	---
Loko	51 (5.3%)	34 (3.4%)	0.038	---	---	---
Korankoh	35 (3.6%)	35 (3.5%)	0.860	---	---	---
Sherbro	17 (1.8%)	19 (1.9%)	0.830	---	---	---
**Primary sources of COVID-19 Information ^1^**						
Mass media	660 (68.3%)	831 (82.5%)	<0.001	0.71	0.54–0.92	0.011
Family or friends	563 (58.2%)	458 (45.5%)	<0.001	---	---	---
National government/Ministry of Health	347 (35.9%)	533 (52.9%)	<0.001	---	---	---
Social media	273 (28.2%)	536 (53.2%)	<0.001	0.74	0.58–0.93	0.011
Health care workers	169 (17.5%)	385 (38.2%)	<0.001	0.44	0.34–0.56	<0.001
Local leaders	114 (11.8%)	176 (17.5%)	<0.001	---	---	---
Employer	15 (1.6%)	69 (6.9%)	<0.001	---	---	---

^1^ Respondents could indicate multiple ethnicities and primary sources of information about COVID-19 vaccines. ^2^ Chi-squared or Fisher’s exact tests were used. ^3^ Wald test was used.

**Table 4 vaccines-11-01385-t004:** Predictors of poor COVID-19 vaccine acceptability (responded no or unsure to the intention of receiving a COVID-19 vaccine) (N = 1974).

	Univariate			Logistic Regression Model		
	Do Not Intend to Be Vaccinated (No/Unsure)	Intend to Be Vaccinated	*p*-Value ^2^	OR	95% CI	*p*-Value ^3^
	N = 535	N = 1439				
**Location**			<0.001			<0.001
Koidu Government Hospital	78 (14.6%)	521 (36.2%)		4.59	1.93–10.90	
Wellbody Clinic	110 (20.6%)	395 (27.4%)		8.74	3.62–21.09	
Sewafe Community Health Center	6 (1.1%)	134 (9.3%)		1.00	---	
Konbayedeh Community Health Center	37 (6.9%)	77 (5.4%)		4.70	1.78–12.42	
Lakka Government Hospital	155 (29.0%)	131 (9.1%)		26.37	10.77–64.58	
Kissy Psychiatric Teaching Hospital	149 (27.9%)	181 (12.6%)		32.90	13.78–78.58	
**Sex**			0.380			
Female	284 (53.1%)	732 (50.9%)		---	---	
Male	251 (46.9%)	707 (49.1%)		---	---	
**Age, categorized (years)**			0.580			0.009
18–24	116 (21.7%)	276 (19.2%)		1.98	1.29–3.06	
25–34	209 (39.1%)	596 (41.4%)		1.36	0.93–1.97	
35–44	139 (26.0%)	366 (25.4%)		1.17	0.79–1.72	
≥45	71 (13.3%)	201 (14.0%)		1.00	---	
**Level of education**			<0.001			<0.001
Did not complete primary	197 (36.8%)	405 (28.1%)		3.16	2.18–4.57	
Completed primary	170 (31.8%)	444 (30.9%)		1.75	1.22–2.50	
Completed secondary	92 (17.2%)	302 (21.0%)		1.51	1.02–2.26	
Post-secondary	76 (14.2%)	288 (20.0%)		1.00	---	
**Ethnicity ^1^**						
Kono	147 (27.5%)	610 (42.4%)	<0.001	---	---	---
Krio	260 (48.6%)	297 (20.6%)	<0.001	2.97	2.20–4.01	<0.001
Mende	99 (18.5%)	250 (17.4%)	0.560	---	---	---
Temne	90 (16.8%)	175 (12.2%)	0.007	---	---	---
Limba	61 (11.4%)	127 (8.8%)	0.083	---	---	---
Fullah	58 (10.8%)	102 (7.1%)	0.007	---	---	---
Mandingo	39 (7.3%)	120 (8.3%)	0.450	---	---	---
Other	29 (5.4%)	74 (5.1%)	0.800	---	---	---
Loko	27 (5.0%)	58 (4.0%)	0.320	---	---	---
Korankoh	18 (3.4%)	52 (3.6%)	0.790	---	---	---
Sherbro	13 (2.4%)	23 (1.6%)	0.220	---	---	---
**Primary sources of COVID-19 Information ^1^**						
Mass media	335 (62.6%)	1156 (80.3%)	<0.001	0.65	0.50–0.85	0.002
Family or friends	357 (66.7%)	664 (46.1%)	<0.001	---	---	---
National government/Ministry of Health	216 (40.4%)	664 (46.1%)	0.022	---	---	---
Social media	163 (30.5%)	646 (44.9%)	<0.001	---	---	---
Health care workers	73 (13.6%)	481 (33.4%)	<0.001	0.31	0.23–0.42	<0.001
Local leaders	77 (14.4%)	213 (14.8%)	0.820	---	---	---
Employer	8 (1.5%)	76 (5.3%)	<0.001	---	---	---

^1^ Respondents could indicate multiple ethnicities and primary sources of information about COVID-19 vaccines. ^2^ Chi-squared or Fisher’s exact tests were used. ^3^ Wald test was used.

## Data Availability

The data presented in this study are available upon request from the corresponding author and pending approval from the PIH Sierra Leone Research Review Committee.

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
