# Peer review of "Attitudes toward COVID-19 Vaccination: Staff and Patient Perspectives at Six Health Facilities in Sierra Leone"

_vaccines, 2023, doi:10.3390/vaccines11081385_

Round 1
Reviewer 1 Report
1. This is a valuable paper to improve the COVID-19 vaccination rate in Sierra Leone.
2. The conclusion of the abstract(Line30-32:Lack of access to COVID-19 vaccines is still an underlying problem in Sierra Leone, however addressing vaccine hesitancy will assist in improving vaccine coverage) is inconsistent with the conclusion of the main body of the paper(Line325-327:This indicated that lack of access to vaccines, rather than lack of willingness to be vaccinated, is the primary barrier to successful COVID-19 vaccine campaigns in Sierra Leone).The conclusions of the paper are not comprehensive enough, such as the reasons for lack of access to COVID-19 vaccines and vaccine hesitancy.More important data should be provided in the abstract to support the conclusions of the paper.
3. It is recommended to divide Figure 1 into A and B. The vertical axis of the left figure should mark out the unit, such as%.
4. Is the statistical analysis in Tables 3 and 4 using single factor analysis or multi factor analysis? It should be noted. If it is a multi factor analysis, it should be noted which factors are controlled.
5. Each factor of “Location;Role at the facility;Sex;Age;Level of education;”should indicate a P-value in Tables 3 and Tables 4.
6. Which factor is used as a reference for the statistical results of”Ethnicity,Primary sources of COVID-19 Information” in Tables 3 and 4?
7. It is necessary to make suggestions on how to improve the COVID-19 vaccination rate in Sierra Leone according to the research results in discussion.
Minor editing of English language required.
Reviewer 2 Report
Nicely written. The fact that the survey was made available to individuals in their native language and also in a format that was read to the individuals and then entered into the computer database didn't exclude participants who desired to fill out the survey but didn't have computer access or knowledge.
The authors did state the limitations in the survey in that only individuals that visited the care facilities were included into the survey. Individuals who visit a health care facilities may be more inclined to seek health care and seek vaccinations.
It would be interesting for a follow up survey of the general populations view on the acceptance and attitude towards vaccination.
Author Response
Thank you for your kind remarks.
Reviewer 3 Report
This is a well written paper deserving publication. I did not see any issues and the paper can be published as is.
Author Response
Thank you for your kind remark.